# Physiological and Molecular Response of *Liriodendron chinense* to Varying Stand Density

**DOI:** 10.3390/plants13040508

**Published:** 2024-02-11

**Authors:** Jun Chen, Ting Li, Jinfeng Cai, Pengfei Yu, Ying Guo

**Affiliations:** 1Co-Innovation Center for Sustainable Forestry in Southern China, College of Forestry, Nanjing Forestry University, Nanjing 210037, China; treeseeds@126.com (J.C.); tinglee@njfu.edu.cn (T.L.); caijinfeng1984@126.com (J.C.); 2Suining County Run Enterprises Investment Co., Ltd., Xuzhou 212100, China; bjseeds@126.com

**Keywords:** stand density, *Liriodendron chinense*, transcription factors, photosynthesis, regulatory mechanism

## Abstract

Stand density affects the potentially superior productivity of forest ecosystems directly by regulating the light and nutrient availability of trees. Understanding how stand density influences the growth and development of trees is crucial for supporting forest management in the context of climate change. We focused on *Liriodendron chinense* in experimental plantations created in 2003, with planting densities ranging from 277 to 10,000 trees per hectare at six plots. The leaf structure and photosynthetic capacity of *L. chinense* changed significantly under different stand densities, which had a negative impact on their biomass (leaf mass) and nutrient (total carbohydrate content) accumulation. Transcriptional differences were observed among samples from plots with different planting densities. The expression of 1784 genes was negatively dependent on stand density, participating mainly in the biological processes of “circadian rhythm”, “carbon metabolism”, and “amino acid biosynthesis”. Furthermore, we identified a photosynthesis-related module and constructed a gene regulatory network to discover that the transcription factors of MYB and bHLH may have important roles in the transcriptional regulation of photosynthesis biosynthesis by activating or repressing the expression of *petA* (*Litul.15G096200*), *psbE* (*Litul.10G033900*), and *petD* (*Litul.17G061600*) at different stand densities. Our study quantified the impact of stand density on tree growth at physiological and molecular levels. Our observations provide theoretical support for plantation establishment of *L. chinense*.

## 1. Introduction

Photosynthesis integrates biochemical reactions in time and space to convert light energy into chemical energy, which represents the main mechanism of production of plant dry matter. The main function of the light-dependent reaction is to absorb and convert light energy, provide an “assimilation force” for the light-independent reaction, and then assimilate CO_2_ into carbohydrates [1,2]. For trees, improved photosynthesis can increase carbon fixation and production of biological materials such as wood and fiber. Research has shown that >90% of a tree’s dry weight is the direct or indirect product of photosynthesis, and that the net photosynthetic rate can be used to approximate the rate of dry-matter production of trees [3]. Therefore, understanding the photosynthetic characteristics of tree species will help to understand the biological traits and ecological roles of these species, including interactions with other organisms and their adaptations to the environment. In this way, appropriate cultivation techniques and measures can be adopted to ensure that the photosynthetic potential of excellent varieties (clones) is utilized fully.

Different planting densities lead to different environmental conditions in the forest stand, which affects the efficiency of light-energy utilization and organic-matter accumulation in plants. In general, after a certain age, as the stand density increases, the net photosynthetic rate of forest trees decreases [4]. Ouyang et al. showed that, in a 5-year-old poplar forest stand, the net photosynthetic rate of a stand density of 625 plants·ha^−1^ was lower than that of a stand of 204 plants·ha^−1^ by 3–5 mg CO_2_·dm^−2^·h^−1^. The main reason for this difference was that the amount of light received by leaves of different densities was significantly different [5]. In addition, due to the obstruction of airflow movement in denser stands, the CO_2_ concentration in the canopy was low. When poplar species carry out photosynthesis rapidly, the net photosynthetic rate of the forest trees is limited due to a lack of CO_2_. Leaves are the main organs of photosynthesis, which are sensitive to environmental changes during their evolution. Studies have shown that the anatomical traits of leaves develop in a direction that is conducive to improving the efficiency of water use. They are affected by sunlight, evapotranspiration, and precipitation, and are expressed as leaf area as well as changes in the thickness and microstructure of leaves [6,7,8]. In addition, photosynthesis is the only source of carbon in plants. The content of carbohydrates (e.g., starch and sugar) in leaves reflects the carbon balance of the tree in photosynthesis (carbon absorption) and growth and development (carbon consumption). However, the effect of forest stand density (FSD) on photosynthetic efficiency and accumulation of photosynthetic products has not been analyzed thoroughly.

*Liriodendron chinense* is a rare relict tree species of the tertiary period, and it is a precious second-level protected plant in China. Currently, it has been widely cultivated due to its rapid growth, high ornamental value, and versatile wood. The release of the genomes of *Liriodendron tulipifera* and *L. chinense* (Hemsl.) Sargent. has aided the molecular breeding and research into genetic improvements [9]. The “transcriptome” refers to the sum of expression products of all genes in an organism [10]. Transcriptome-related studies based on systems biology have provided new insights into the regulatory mechanisms of gene expression related to environmental responses [11,12,13]. During the growth of forest trees, cells in each tissue exhibit significant regulatory flexibility. This feature allows them to survive in response to sudden environmental changes by initiating gene-expression programs that help regulate cellular metabolism to adapt to new conditions [14,15]. In recent years, several studies related to the transcriptome of *L. tulipifera* (e.g., excavation of their cold-resistance genes; response to flooding stress and its regulatory mechanism; petal-coloring mechanism) have been conducted [16,17,18]. Changes in stand density causes changes in the growth environment of forest trees, such as changes in water, heat, and light conditions [19]. Nevertheless, the impact of different stand densities on the transcription levels of cultivated *L. chinense* is still unclear.

Herein, we took *L. chinense* standard tree in six afforestation density plots as the research object to explore the effect of row spacing on the leaf microstructure, biomass, photosynthetic characteristics, photosynthetic-product content, and nutritional element content in leaves. This was done to determine the appropriate afforestation density and provide a scientific basis for the management strategy of *L. chinense* plantations. Furthermore, functional leaves were collected for transcriptome sequencing to explore the effect of stand density on gene expression. In this way, we aimed to understand more deeply the response of gene expression to environmental changes.

## 2. Results

### 2.1. Impact of FSD on the Leaf Anatomy of L. chinense

Leaves are essential organs for the synthesis of organic compounds through photosynthesis. Their morphological characteristics and basic tissue structure are closely related to the adaptability of trees to FSD. Microscopic observations revealed that the upper epidermis and lower epidermis of *L. chinense* leaves consisted of a single layer of cells (Figure 1). Upper-epidermal cells had a consistent and approximately hexagonal-to-elliptical shape. Lower-epidermal cells exhibited a more diverse, irregular, and elliptical shape. The average thickness of the cuticle layer on the upper surface and lower surface of leaves from the six stand densities was 3.44 μm and 3.36 μm, respectively. The upper-epidermal cells had a greater length (mean, 32.27 μm) compared with that of lower-epidermal cells (mean, 28.78 μm).

There were significant differences in the microscopic structural characteristics of the leaf epidermis among samples from different FSDs (*p* < 0.05) (Table 1). In groups with higher FSD (density groups 1–4), the thickness of the upper-epidermal cuticle layer was significantly greater than that of groups with lower FSD (density groups 5–6), but a similar phenomenon was not observed in the thickness of the lower-epidermal cuticle layer. Samples from groups with higher FSD had longer upper-epidermal cells, with the length of upper-epidermal cells decreasing sequentially with decreasing FSD. Cell width also varied among samples from different groups, with the upper- and lower-epidermal cell widths of samples from density group 4 being significantly greater than those of other groups, and a small number of bilayered epidermal cells were observed (Figure 1). In addition, the length:width ratio of upper- and lower-epidermal cells from samples from density group 6, with a FSD of 6 m × 6 m, was significantly higher than that of the other samples, and measured 2.57 and 2.42, respectively.

### 2.2. Influence of FSD on Biomass and Nutrient Accumulation in L. chinense

Analysis of the mass of a single leaf revealed significant differences in the fresh weight and dry weight of *L. chinense* leaves grown under different FSDs (*p* < 0.05), with mean values of 2.71 g and 1.05 g, respectively. The fresh weight and dry weight of leaves increased as the FSD decreased (Figure 2a,b). Samples from density group 6 had significantly higher fresh weight and dry weight of individual leaves, with mean values of 4.13 g and 1.96 g, respectively. Samples from density group 1 had the lowest fresh weight and dry weight among all groups, measuring 1.94 g and 0.66 g, respectively. There were no significant differences in fresh weight or dry weight between single leaves in density groups 2 to 4. The variation trends of leaf area and specific leaf area in samples with different densities were consistent (Figure 2c,d), roughly showing an increasing trend with increasing density.

Analyses and comparison of photosynthetic-product contents in *L. chinense* leaf samples revealed significant differences in the content of total sugar and total starch in leaves from FSDs (*p* < 0.05). The total sugar content in *L. chinense* leaves was approximately 225–265 mg·g^−1^, and total starch content was approximately 175–225 mg·g^−1^ (Figure 3a,b). Total sugar content decreased gradually with decreasing FSD, except for a slight increase in total sugar content in samples from density groups 5 and 6. Samples from density group 1 had significantly higher total sugar content than leaves from other density groups. In contrast, the trend for total starch content was the opposite, with samples from density groups 5 and 6 having significantly higher total starch content compared with those in leaves from other density groups.

In *L. chinense* samples from different FSDs, the change trend of total carbon was similar to that of total sugar, but samples from density group 5 contained the highest total carbon content (525.11 mg·g^−1^), whereas samples from density group 4 had the lowest total carbon content (451.31 mg·g^−1^), revealing a significant difference between the two (Figure 3c). Likewise, the highest total nitrogen content was measured in samples from density group 5 (25.48 mg·g^−1^), significantly higher than the total nitrogen content in leaves from other density groups (Figure 3d). Total phosphorus content in *L. chinense* leaves was closely related to density. Leaves from high-density groups (samples from density groups 1 and 2) had significantly higher total phosphorus content (~1.4 mg·g^−1^), whereas leaves from low-density groups (samples from density groups 5 and 6) had significantly lower total phosphorus content (~1.0 mg·g^−1^) (Figure 3e). The lowest and highest total potassium content was detected in samples from density groups 1 and 2 and measured 1.62 mg·g^−1^ and 3.15 mg·g^−1^, respectively. Total potassium content decreased with increasing FSD in other samples (Figure 3f).

### 2.3. Comparison of Photosynthetic Characteristics in L. chinense under Different FSDs

Photosynthesis is the process of converting light energy into chemical stored energy. Photosynthetic physiological parameters are used commonly to characterize plant growth and environmental adaptability. Analysis of variance indicated significant differences in various photosynthetic parameters of *L. chinense* planted under different FSDs (Figure 4).

The mean value of the instantaneous net photosynthetic rate of *L. chinense* leaves from different groups was 9.13 μmol·m^−2^·s^−1^. The sample size ranking of the instantaneous net photosynthetic rate (in μmol·m^−2^·s^−1^) was density 5 (15.66) > density 6 (12.30) > density 4 (10.62) > density 3 (7.92) > density 2 (4.5) > density 1 (3.8) (Figure 4a). Samples from density group 1 had significantly higher stomatal conductance compared to that of other density group samples. The lowest stomatal conductance was in samples from density group 6 (Figure 4b), measuring 410.8 and 261 μmol·mol^−1^. The stomatal conductance was similar in samples from density groups 2 to 5, with a total mean stomatal conductance of 313 μmol·mol^−1^ for all six FSD samples.

Intercellular CO_2_ concentration showed significant differences among samples with different FSDs (Figure 4c), but did not exhibit a clear trend. A total mean intercellular CO_2_ concentration of 171.13 μmol·m^−2^·s^−1^ was documented for all samples. Samples from density group 6 had the highest intercellular CO_2_ concentration (246.2 μmol·m^−2^·s^−1^), approximately 3.29 times higher than that of samples from density group 2 (which had the lowest intercellular CO_2_ concentration).

The transpiration rates of *L. chinense* leaves exhibited significant differences between two groups among the six FSD samples (Figure 4d). Specifically, samples from lower FSDs (density groups 4–6) had significantly higher transpiration rates than those from higher FSDs (density groups 1–3), with mean values of 4.82 and 2.37 μmol·m^−2^·s^−1^, respectively. Vapor-pressure deficit and water-use efficiency showed significant differences among samples with different FSDs (Figure 4e,f), but they exhibited fluctuations without a clear trend, with mean values of 2.19 kPa and 2.51 μmol·mol^−1^, respectively. Samples from density group 2 (2.80 kPa) and density group 4 had the highest and lowest vapor-pressure deficits, respectively. Simultaneously, samples from density group 5 (3.42 μmol·mol^−1^) and density group 1 (1.78 μmol·mol^−1^) had the highest and lowest water-use efficiencies, respectively.

### 2.4. Differential Transcriptome Analyses of Different FSD Samples

PCA was conducted using the expression data from 18 sequencing samples to explore the relationships between samples through dimensionality reduction (Figure 5a). The PCA score plot demonstrated separation between samples from different FSDs, with biological replicates clustering closely together within each group and no outliers. PC1 and PC2 accounted collectively for 48.9% of the total variance in gene expression in the dataset, and the sample distribution along the PC1 axis was notable. Samples from FSDs 1–4 were situated within the first and third quadrants of the PC1 negative axis. Samples from FSDs 5 and 6 were placed in the second and fourth quadrants of the PC1 positive axis. PC1 contributed to 30.6% of the total variance. In the PCA score plot, samples from densities 1, 3, and 4 exhibited closely clustered distributions, indicating similarity in their gene-expression patterns. The correlation heatmap visually illustrated the correlation between any two samples by calculating the Pearson correlation coefficient for the expression between pairs of samples (Figure 5b). The biological replicates within each FSD exhibited high correlations (all > 0.99), thereby demonstrating robust reproducibility within each group, which supported subsequent inter-group differential analysis of expression. Furthermore, strong correlation was observed between samples from density 3 and 4, as well as between samples from density 5 and 6, which signified the influence of different FSDs on gene expression in the functional leaves of trees.

Following the filtration of genes with expression (FPKM) < 1, the remaining genes were subjected to STEM analysis to explore trends in gene expression as FSD decreased (Figure 5c). The 18 transcriptome datasets were categorized into 20 expression profiles, with seven profiles exhibiting significant enrichment (*p* < 0.05). Profile 0 contained 1906 genes displaying decreased expression as FSD decreased, whereas Profile 19 encompassed 1784 genes with the converse expression trend: gene expression increased with decreasing FSD. Using the KEGG database, analyses of signaling-pathway enrichment in genes in Profile 0 (which depicted a decrease in expression with decreasing FSD) revealed enrichment in environmental-information processing and metabolic pathways. These included pathways such as “plant hormone signal transduction” (ko04075), “ABC transporters” (ko02010), “starch and sucrose metabolism” (ko00500), “taurine and hypotaurine metabolism” (ko00430), and “pentose and glucuronate interconversions” (ko00040). Genes in Profile 19 (which showed increased expression with decreasing FSD) were enriched primarily in four categories of biological metabolic pathways: genetic-information processing, metabolism, organismal systems, and environmental-information processing. These pathways encompassed “plant circadian rhythms” (ko04712), “protein processing in the endoplasmic reticulum” (ko04141), “ribosome” (ko03010), “carbon metabolism” (ko01200), and “amino acid biosynthesis” (ko01230).

To examine the impact of stand density on gene expression in the functional leaves of *L. chinense*, we used the sample with the highest FSD (density 1) as the control and conducted differential analysis step-by-step (Figure 5d). The comparison between the sample with the highest FSD (density 1) and the sample with the lowest FSD (density 6) revealed the largest number of differentially expressed genes (DEGs), with 2556 DEGs having upregulated expression and 2617 DEGs having downregulated expression in each group. The fewest DEGs were found in the comparison between density 1 and density 3, with 716 DEGs with upregulated expression and 530 DEGs with downregulated expression. Comparison of density 1 with density 2, density 4, and density 5 resulted in 4498, 3006, and 3593 DEGs, respectively. In general, except for density 2, the number of DEGs increased gradually with higher FSD. In total, 9452 DEGs were obtained from the five comparison groups, including 93 genes related to photosynthesis.

### 2.5. Analyses of the Co-Expression Network of Photosynthesis-Related Genes

After filtering low-expression data (FPKM > 5), the transcriptome data of the 18 sample groups were used for analyses of a gene co-expression network. We discovered that 15,742 genes were grouped into 20 modules with highly correlated gene expressions within each module. The module eigengenes were associated with photosynthetic parameters (net photosynthetic rate), photosynthetic products (total sugar content and total starch content), nutritional elements (total contents of carbon, nitrogen, phosphorus, and potassium), and growth traits (leaf fresh weight) (Figure 6a).

In the black module, gene expression was significantly positively correlated with the net photosynthetic rate, total starch content, total carbon content, and total carbon content (*p* < 0.05). Analyses of signaling-pathway enrichment of 830 genes in the black module showed that module genes were enriched mainly in pathways related to photosynthesis, carbon metabolism, and carbon fixation in photosynthetic organisms (Figure 6b).

Based on the weighted gene co-expression network constructed with the top-100 connected genes in the black module (Figure 6c), several MYB genes related to TFs (*Litul.02G032000*, *Litul.09G101400*, *Litul.02G031800*, *Litul.02G031900*) and one bHLH TF gene (*Litul.18G056500*) were identified. Research has indicated that bHLH is involved in the regulation of photosynthesis-related genes in gooseneck loosestrife (*Lysimachia clethroides*), such as *petA* (*Litul.15G096200*), *psbE* (*Litul.10G033900*), and *petD* (*Litul.17G061600*). Expression of the gene for ribosomal protein S2 (*Litul.11G046700*) involved in the metabolic pathways of starch and sucrose may be regulated by TFs related to MYB.

## 3. Discussion

Afforestation in artificial forests is a crucial component of global ecosystems. China boasts the largest artificial forest area worldwide, covering ~80 million hectares [20]. The development of artificial forests plays a pivotal part in carbon sequestration. During afforestation, the planting density of different tree species is of paramount importance for afforestation quality and the healthy development of forest stands [21]. Planting density forms the foundation of forest stand structure, and alterations to this setting can affect environmental factors within the forest stand, such as light, temperature, and moisture, consequently leading to changes in the growth status of trees [22,23,24,25]. Excessive density can result in the wastage of land resources. Insufficient density may limit water and thermal conditions, leading to poor growth. Therefore, determining an appropriate planting density is crucial for ensuring population structure and resource allocation.

The traits of leaf growth are the horizontal characteristics of leaves that reflect the adaptation of a tree to its environment. They play a crucial part in assessing FSD suitability. Studies have shown that FSD affects leaf biomass significantly [26,27]. Our results indicated that in stands of density 5 (5 m × 5 m) and 6 (6 m × 6 m), the accumulation of leaf biomass in *L. chinense* trees was the highest, whereas the single-leaf area was significantly lower than that in samples from the density 1 (1 m × 1 m) stand (Figure 2). In relatively low-density *L. chinense* plantations, the specific leaf area of *L. chinense* was smaller, whereas in stands with relatively high density, the specific leaf area of *L. chinense* was larger. Perhaps due to the high correlation between specific leaf area and light level, which decreases with increasing light intensity, so trees grown under low-density planting usually have thicker leaves with lower specific leaf area [28]. Consistent with this, research on *Eucalyptus* clones has found that the leaf area index increased with planting density, while individual biomass increment decreased with increasing density [29]. The average tree diameter of Siberian spruce (*Picea obovata* Ledeb.) demonstrated negative dependence on stand density, as well as mean tree-ring width and latewood width decreased with rising stand density [25]. Research found that stand density had a direct impact on soil moisture, and the availability of water is one of the main limiting factors for tree growth [30]. Low density stands displayed substantially higher long-term average growth than higher density stands in response to greater long-term moisture availability. In the context of climate change, reducing stand density is considered an effective way to mitigate the negative impact of increasingly severe drought [31].

*L. chinense* exhibited significant differences in photosynthetic characteristics at different FSDs, and changes in instantaneous photosynthetic parameters reflected the response to site conditions. Consistent with research on soybeans, lower FSDs provide better light penetration and ventilation, reduce competition among individuals, and promote an increase in the net photosynthetic rate [32]. These phenomena, in turn, enhance the leaf weight, chlorophyll content, and photosynthetic-product accumulation. The trends in water utilization efficiency and transpiration rate among different FSDs were consistent with the net photosynthetic rate. That is, lower-FSD samples (density 4–6) were significantly higher than samples from higher-FSDs (Figure 4). For example, the net photosynthetic rate in density 5 samples was significantly higher than in other samples (Figure 4). Photosynthesis is the primary source of carbon for plants, providing the necessary energy for growth and the foundation for material production [33]. The content of carbohydrates such as starch and total sugars reflects the carbon balance in trees between photosynthesis (carbon assimilation) and growth and development (carbon consumption). As FSD decreases, canopy light penetration increases, leading to enhanced capture of light energy by leaves. This action boosts the net photosynthetic rate and creates favorable conditions for dry-matter accumulation in leaves [34,35]. However, high density may also have a positive impact on plant growth by reducing wind speed to reduce transpiration or through hydraulic redistribution [36].

Research on *Eucalyptus robusta* has shown that higher planting densities can lead to restricted growth space for trees, making them more susceptible to biotic and abiotic stressors [37]. The results of the present study are consistent with those findings. Our analyses revealed that the expression of genes related to plant-defense pathways increased in high-density samples, such as the genes involved in “glutathione metabolism” (ko00480), “biosynthesis of secondary metabolites” (ko00999), and “flavonoid biosynthesis” (ko00941) pathways. These data are consistent with findings from research on Poplar species that showed upregulation of expression of the TFs involved in stress response in high-density plant leaves [38]. Weighted gene co-expression network analysis is a systems-biology method used to assess patterns of correlation of gene expression in multiple samples [39,40]. Based on analyses of a gene co-expression network, gene expression in the black module was found to have a significant positive correlation with the net photosynthetic rate, total starch content, total carbon content, and total carbon content (Figure 6a). Several genes encoding MYB TFs were identified in the constructed co-expression network (Figure 6c). The MYB family is one of the largest TF families in plants. It is involved in regulating various plant-specific biological processes, such as photomorphogenesis, secondary metabolism, as well as biotic and abiotic stress responses [41,42]. MYB112 is a negative regulator of photomorphogenesis in *Arabidopsis* species [43]. It can interact directly with the core repressor phytochrome-interacting factor 4 (PIF4) protein and enhance its transcriptional activation of downstream target genes, such as *YUCCA8*, *IAA19*, and *IAA29*. Photosystem II (PSII) is an essential protein complex located on the thylakoid membrane of oxygenic photosynthetic organisms. It utilizes light energy to split water into protons and electrons to release oxygen. We found that bHLH TFs were involved in regulating the expression of several genes encoding the core subunits of PSII, such as *psbC* and *psbE* (Figure 6c). In research on *Pinus koraiensis*, expression of the TFs MYB, AP2-ERF, and bHLH was found to be upregulated specifically during light stress, whereas genes related to PSII core-complex proteins [44], such as *PsbA*, *PsbB*, *PsbC*, *PsbS*, *PsbQ*, and *PsbZ*, exhibited opposite trends.

## 4. Materials and Methods

### 4.1. Overview of the Experimental Site

The experimental site was located in the teaching and research base of Nanjing Forestry University in Xiashu Forest Farm, Jiangsu, China (119°13′ E, 32°7′ N). It is situated in a subtropical monsoon climate zone and has a mild and humid climate. The region experiences four distinct seasons, with most rainfall occurring from June to September. The average annual rainfall is 1105 mm, and the mean annual temperature is 15.2 °C. The frost-free period lasts for 229 days each year. The soil pH ranges from 4.0 to 5.0, and the soil type is yellow–brown.

### 4.2. Experimental Materials and Design

The experimental forest was planted with 1-year-old *L. chinense* seedlings in March 2003. A randomized block experimental design was used with six treatments and three replicates. The six treatments represented different planting densities: D1, 1 m × 1 m (10,000 plants per hectare); D2, 2 m × 2 m (2500 plants per hectare); D3, 3 m × 3 m (1,111 plants per hectare); D4, 4 m × 4 m (625 plants per hectare); D5, 5 m × 5 m (400 plants per hectare) and D6, 6 m × 6 m (277 plants per hectare). Eighteen plots were established, each measuring 20 m × 30 m. After afforestation, cleaning of the undergrowth and shrub was undertaken annually as needed. The average tree height is about 17 m and the average diameter at breast height is about 20 cm. Using a stratified sampling method, three standard trees were selected from each density for various parameter measurements. A 10-m-high platform was built around standard trees for sampling and photosynthetic testing.

### 4.3. Observation of Microscopic Structure

Three healthy leaves representing the overall crown growth were selected from the middle of the standard treetops of the canopy. They were placed in a 50% FAA (mixture of formaldehyde, acetic acid, and ethanol) fixative solution for 24 h at room temperature. Samples were dehydrated in a graded series of ethanol solutions and embedded in paraffin wax using an embedding machine. Then, tissue blocks were sectioned on a paraffin microtome, and the thickness of each section was 4 μm. Sections were stained with toluidine blue. A MIDI scanner (Pannoramic™; 3DHistech, Budapest, Hungary) was used for imaging. CaseViewer (3DHistech) was used to visualize and measure imaging results.

### 4.4. Measurement of Leaf Traits

Three branches (length = 1 m) were cut randomly from the middle of the standard treetops of the canopy. All mature leaves were imaged using a leaf area meter (YMJ-B; Top Cloud-ARGI, Hangzhou, China) to calculate the average leaf area. After imaging, mature leaves were collected and weighed using an analytical balance to determine the weight of a fresh leaf. Subsequently, leaves were placed in an oven at 70 °C, and after 48 h, they were weighed to obtain the weight of a dry leaf. The ratio of leaf area per leaf mass was defined as “specific leaf area”.

### 4.5. Determination of the Nutritional Elements in Leaves

During the growing season (July), 8–10 upper leaves from standard tree plants were collected and mixed with three replicates for each source. Samples were placed in envelopes, brought to the laboratory, dried to a constant weight at 65 °C, and then ground for measurement of various parameters.

Determination of soluble-sugar content was carried out using the anthrone colorimetric method. Starch content was determined using the method based on acid hydrolysis–dinitrosalicylic acid. Total non-structural carbohydrates represent the sum of soluble sugar and starch content. The content of total carbon was determined using the method based on oxidation of potassium dichromate with external heating. Nitrogen content was ascertained using the method of Kjeldahl digestion and diffusion. Contents of phosphorus and potassium were determined through digestion by nitric acid followed by analysis with inductively coupled plasma-atomic emission spectroscopy.

### 4.6. Measurement of Photosynthetic Parameters

In August 2021, on a clear and windless day following heavy rain with the day average temperature was 32 °C, measurements were conducted under natural conditions using a portable photosynthesis system (CIRAS-3; PP-Systems, Amesbury, MA, USA). We randomly selected a branch of the canopy crown on the 10 m high platform, and its 5th and 6th mature leaves at the base were used for photosynthesis detection. Measurements were taken between 8 am and 11 am, and the light source of the photosynthesis instrument is used to ensure consistency in light intensity, with a light intensity set at 1200 μmol·m^−2^·s^−1^. The key photosynthetic parameters were the: instantaneous net photosynthetic rate (Pn); stomatal conductance (Gs); intercellular CO_2_ concentration (Ci); transpiration rate (Tr); vapor pressure deficit (VPD); water utilization efficiency (WUE). The above parameters are displayed and recorded during the measurement.

### 4.7. Sequencing and Analyses of the Transcriptome

Three standard trees were selected for each of the six FSDs. Three leaves were collected from each standard tree in the southeast, northwest, and southwest directions, mixed to create one sample, and placed immediately in liquid nitrogen. Then, samples were stored at −80 °C. Three biological replicate samples were collected for each FSD, resulting in 18 samples for transcriptome sequencing.

Total RNA was extracted from each sample using the TRIzol™ Reagent kit (Invitrogen, Carlsbad, CA, USA). RNA quality was assessed using a bioanalyzer (2100 series; Agilent Technologies, Palo Alto, CA, USA). RNA integrity was confirmed by RNase-free agarose gel electrophoresis. After extraction of total RNA, eukaryotic messenger (m) RNA was enriched using Oligo(dT) beads. Prokaryotic mRNA was removed using the Ribo-Zero™ Magnetic kit (Epicentre, Madison, WI, USA). Enriched mRNA was chopped into short fragments using a fragmentation buffer and reverse-transcribed into complementary (c)DNA using random primers. After assessment of RNA quality, purification, and fragmentation of mRNA, 18 cDNA libraries were constructed. Raw data were obtained through paired-end sequencing using the HiSeq™ 2500 system (Illumina, San Diego, CA, USA).

To obtain high-quality reads, Trimmomatic (version 0.39) was used to remove adapter sequences from the raw transcriptome sequencing reads. Processed reads were mapped to the reference genome of *L. chinense* using Bowtie2 (version 2.3.0) [9]. Fragments per kilobase per million mapped reads (FPKM) were used to standardize gene expression. These FPKM data were utilized for revealing relationships between samples through principal component analysis (PCA) and hierarchical clustering analysis. FPKM data were used directly to estimate differentially expressed genes (DEGs) between samples, with a false discovery rate (FDR)  <  0.05 and |log2 fold change (FC)| > 1 as the thresholds for identifying significant DEGs. Classification of gene function and signaling-pathways of mRNA sequences was undertaken using BLASTx against databases such as Swiss-Prot (www.uniprot.org/, URL (accessed on 15 May 2022)), Gene Ontology (GO; www.geneontology.org/, URL (accessed on 15 May 2022)), and Kyoto Encyclopedia of Genes and Genomes (KEGG; www.genome.jp/kegg/, URL (accessed on 15 May 2022)). Trends in gene expression associated with FSD were obtained using Short Time-series Expression Miner (STEM) software. Gene set enrichment analysis for DEGs between various comparison groups was conducted using software at www.omicsmart.com/ (URL (accessed on 22 May 2022)).

DEGs and related signaling pathways associated with the light-dependent reactions of photosynthesis were identified based on annotations in the KEGG database and GO database. To explore the regulatory mechanism of photosynthesis, MEME Suite (version 4.9.0) (FIMO threshold, 1 × 10^−6^) was used to identify potential transcription factors (TFs) that may target these genes. Transcriptome data were utilized to obtain the expression profiles of these photosynthesis-related genes and TFs. Then, co-expression analysis (Pearson correlation) was undertaken to detect gene expression-related networks; the criteria for gene selection was *p* < 0.05 and |r| > 0.80. A gene regulatory network was generated using Cytoscape (version 3.7.1; https://cytoscape.org/, URL (accessed on 26 July 2022)).

## 5. Conclusions

We analyzed the anatomical structure, growth characteristics, photosynthetic traits, photosynthetic products, and nutritional elements of *L. chinense* cultivated under different tree densities. The lower tree densities (density 5 and density 6) contributed to increased capture of light energy by leaves, thus enhancing the photosynthetic efficiency and promoting biomass accumulation. Furthermore, analyses of transcriptome data revealed multiple TFs, such as MYB, bHLH, bZIP, Dof, GeBP, ERF, and NAC, participate in the regulation of photosynthesis, starch metabolism, sucrose metabolism, and other pathways. Our data provide a scientific basis for the precise cultivation of *L. chinense* plantations.

## Figures and Tables

**Figure 1 plants-13-00508-f001:**
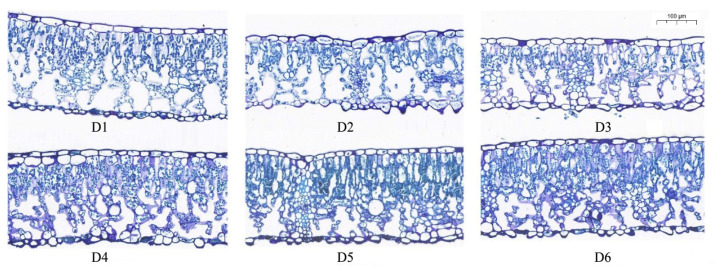
Cross-sectional microstructure of *L. chinense* leaves. D1 to D6 represent samples with different planting densities: D1, 1 m × 1 m (10,000 plants per hectare); D2, 2 m × 2 m (2500 plants per hectare); D3, 3 m × 3 m (1111 plants per hectare); D4, 4 m × 4 m (625 plants per hectare); D5, 5 m × 5 m (400 plants per hectare); D6, 6 m × 6 m (277 plants per hectare).

**Figure 2 plants-13-00508-f002:**
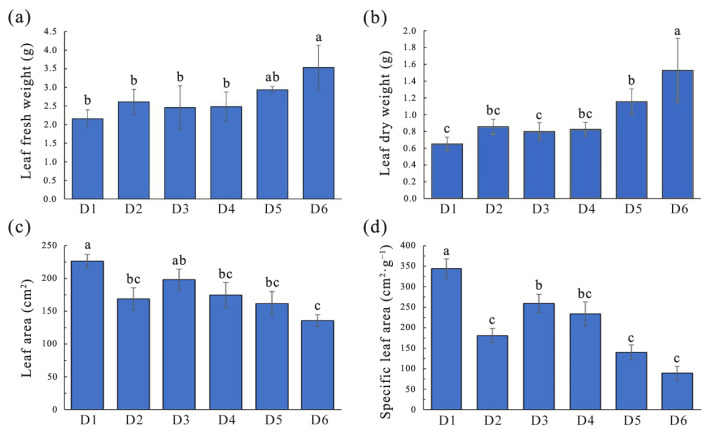
Differences in leaf traits of *L. chinense* under different forest stand densities. (**a**) leaf fresh weight; (**b**) leaf dry weight; (**c**) leaf area; (**d**) specific leaf area. Lowercase letters represent significant differences (*p* < 0.05) between samples.

**Figure 3 plants-13-00508-f003:**
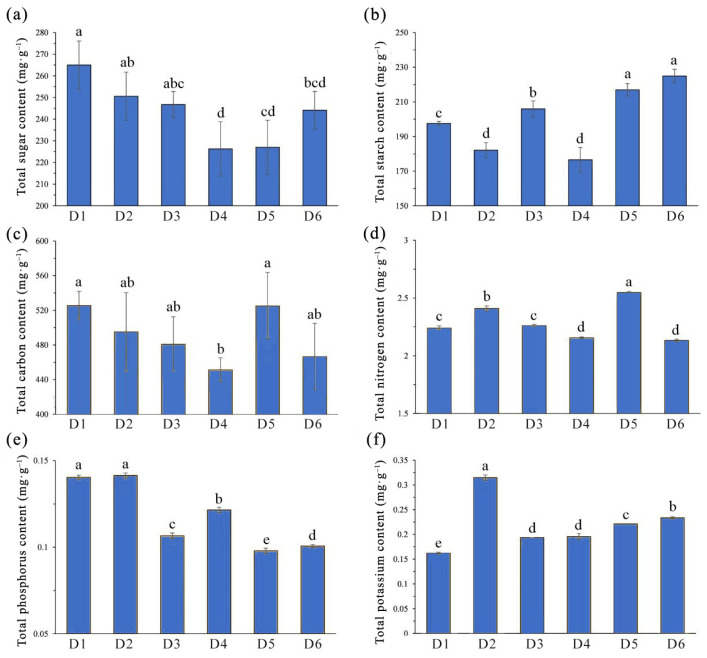
Differences in nutrient content of *L. chinense* under different forest stand densities. (**a**) total sugar content; (**b**) total starch content; (**c**) total carbon content; (**d**) total nitrogen content; (**e**) total phosphorus content; (**f**) total potassium content. Lowercase letters represent significant differences (*p* < 0.05) between samples.

**Figure 4 plants-13-00508-f004:**
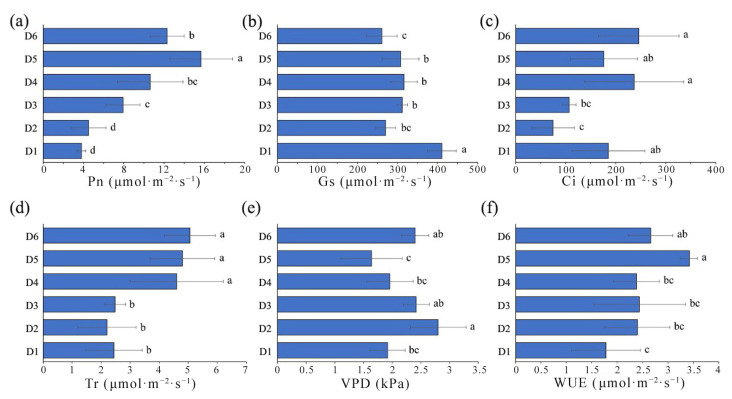
Differences in the photosynthetic parameters of *L. chinense* under different forest stand densities. (**a**) instantaneous net photosynthetic rate (Pn); (**b**) stomatal conductance (Gs); (**c**) intercellular CO_2_ concentration (Ci); (**d**) transpiration rate (Tr); (**e**) vapor-pressure deficit (VPD); (**f**) water utilization efficiency (WUE). Lowercase letters represent significant differences (*p* < 0.05) between samples.

**Figure 5 plants-13-00508-f005:**
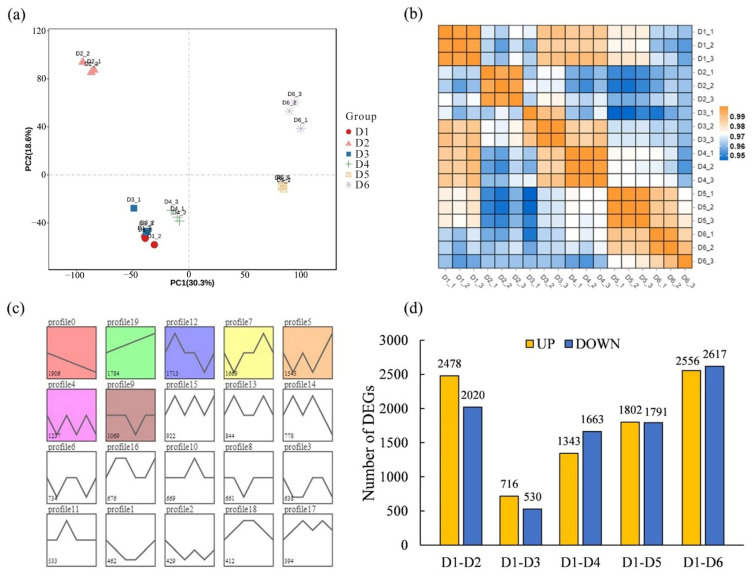
Gene expression in *L. chinense* under different forest stand densities. (**a**,**b**) principal component analysis and cluster analysis of gene expression; (**c**) short time-series expression miner (STEM) analysis of gene expression; (**d**) gene differential expression analysis using D1 sample as control.

**Figure 6 plants-13-00508-f006:**
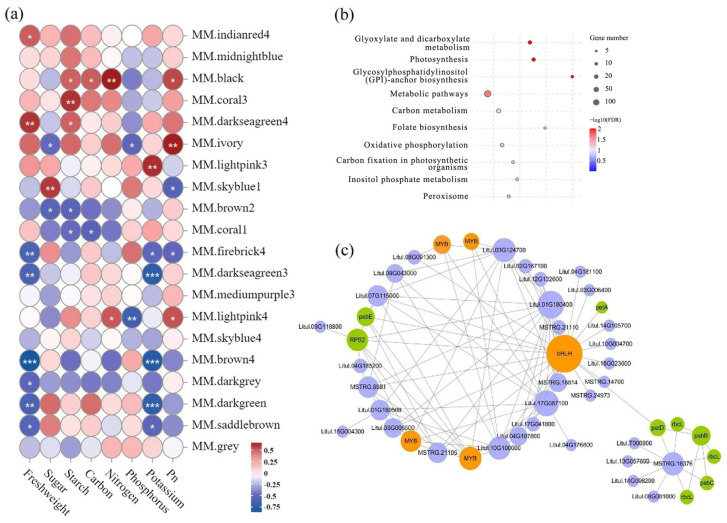
Construction of a co-expression network for photosynthesis-related genes. (**a**) Correlation analysis between module eigenvalues and photosynthetic related traits. The red circles represent positive correlation, while the blue circles represent negative correlation. The star represents a significant correlation (*: *p* < 0.05, **: *p* < 0.01; ***: *p* < 0.001). (**b**) KEGG enrichment analysis of genes within the black module. (**c**) Analyses of the Co-expression network. The green circles represent photosynthesis-related genes, while the orange circles represent transcription factor coding genes.

**Table 1 plants-13-00508-t001:** Main indices of the cross-section microstructure of the leaf epidermis of *L. chinense*.

Parts	Indicator (μm)	Samples						Mean
		D1	D2	D3	D4	D5	D6	
Upper epidermis	Stratum corneum thickness	3.70 a	3.80 a	3.50 a	3.73 a	2.97 b	2.93 b	3.44
	Cell length	36.53 a	36.17 a	31.73 ab	31.70 ab	28.73 b	28.77 b	32.27
	Cell width	14.33 a	13.53 b	13.67 ab	14.60 a	10.33 c	9.93 c	12.73
	Cell length/width	2.54 bc	2.67 ab	2.33 cd	2.17 d	2.78 ab	2.90 a	2.57
Lower epidermis	Stratum corneum thickness	2.93 c	2.27 d	3.27 c	4.27 a	3.93 ab	3.50 bc	3.36
	Cell length	30.67 a	27.63 a	27.57 a	29.63 a	27.60 a	29.57 a	28.78
	Cell width	13.27 ab	12.17 b	13.20 ab	13.97 a	9.57 c	10.47 c	12.11
	Cell length/width	2.30 b	2.28 b	2.09 b	2.12 b	2.88 a	2.83 a	2.42

Lowercase letters represent significant differences (*p* < 0.05) between samples.

## Data Availability

The RNA sequencing data generated as part of this study are deposited in the NCBI Sequence Read Archive (SRA), accession BioProject number PRJNA903548.

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
