# Peer review of "Physiological and Molecular Response of Liriodendron chinense to Varying Stand Density"

_plants, 2024, doi:10.3390/plants13040508_

Round 1

Reviewer 1 Report

Comments and Suggestions for Authors

The investigation is carefully designed, implemented, and reported. It appears that the applied methods are very sophisticated, and obviously appropriate for the purpose.

This reviewer has some questions regarding the application of the results, as well as statements related to environmental conditions, as well as their effect on applications.

The plant density varies by a factor of 33. The photosynthetic efficiency varies by a factor of 3.5. Does this mean that the photosynthetic capacity per hectare is close to ten times in the case of the highest plantation density?

Interpretation of the effect of plant density in physical growing conditions requires elaboration, possibly along with references. Indeed, dense vegetation reduces air circulation. Does it not also provide shadow, then? This reviewer would suspect both of these effects might retain moisture in the ground, and possibly also reduce excess temperatures within the vegetation zone. Many of the statements in the Discussion appear contrary to the above. What is the truth?

Many markings in the Figures are too small for this reviewer to read.

This reviewer would think hm^2 would refer to 0.01 hectares.

Comments on the Quality of English Language

Please check some expressions.

Author Response

RESPONSE TO COMMENTS

Reviewer #1:

The investigation is carefully designed, implemented, and reported. It appears that the applied methods are very sophisticated, and obviously appropriate for the purpose. This reviewer has some questions regarding the application of the results, as well as statements related to environmental conditions, as well as their effect on applications.

Response: Thanks for the comments. We have made some revisions based on these suggestions.

  1. The plant density varies by a factor of 33. The photosynthetic efficiency varies by a factor of 3.5. Does this mean that the photosynthetic capacity per hectare is close to ten times in the case of the highest plantation density?

Response: The results of the photosynthesis determination indicated that there was a difference of about ten times in instantaneous net photosynthetic rate between samples from density 1 and density 6 sites (Line: 173-175). However, key factors such as temperature, light intensity, and carbon dioxide concentration change daily over time, dynamically regulating plant photosynthetic efficiency. Therefore, we believe that a ten-fold difference of photosynthetic capacity per hectare is only an instantaneous result, and there may be more complex situations in production practice.

  1. Interpretation of the effect of plant density in physical growing conditions requires elaboration, possibly along with references. Indeed, dense vegetation reduces air circulation. Does it not also provide shadow, then? This reviewer would suspect both of these effects might retain moisture in the ground, and possibly also reduce excess temperatures within the vegetation zone. Many of the statements in the Discussion appear contrary to the above. What is the truth?

Response: We believe that the reviewer’s questioning is reasonable, as studies have shown that the interaction between stand density and growth environment is complex. Research found that stand density had a direct impact on soil moisture, and the availability of water is one of the main factors limiting tree growth. Low density stands displayed substantially higher long-term average growth than higher density stands in response to greater long-term moisture availability. For example, the studies found that individual biomass increment of Eucalyptus clones decreased with increasing density and average tree diameter of Siberian spruce demonstrated negative dependence on stand density (Line: 312-322). Meanwhile, high density may also have a positive impact on plant growth by reducing wind speed to reduce transpiration or through hydraulic redistribution (Line: 341-342).

  1. Many markings in the Figures are too small for this reviewer to read.

Response: According to the reviewer's suggestions, we have updated the figures in the paper (see Figure 2 to Figure 5).

  1. This reviewer would think hm^2 would refer to 0.01 hectares.

Response: We use the abbreviation “ha” instead of “hm2” to represent hectares (Line: 49-50).

Reviewer 2 Report

Comments and Suggestions for Authors

The leaf gas exchange measurements are useless unless accompanied by indications of the PPFD, Tleaf, CO2, and VPD during the measurements.  Was a lamp used, or natural light?  The Ci values as high as 410 umol mol-1 suggest high CO2 during the measurements.  A careful description of how measured leaves were chosen is also needed - where in the canopy?  It seems like the sampling of leaves for epidermal structure differed from that for SLW, and the leaf gas exchange sampling is thus totally unclear.  

Comments on the Quality of English Language

Overall, fairly good.  I have no idea what "ecological learning ability" means.

Author Response

RESPONSE TO COMMENTS

Reviewer #2:

  1. The leaf gas exchange measurements are useless unless accompanied by indications of the PPFD, Tleaf, CO2, and VPD during the measurements. 

Response: We fully agree with the reviewer’s statement, while we did not measure leaf gas exchange in this paper. Here, we speculated that relatively lower FSDs favored the efficient use of solar energy and gas exchange within L. chinense populations by analyzing the trends in water utilization efficiency, transpiration rate and net photosynthetic rate among different FSDs (Line: 328-333).

  1. Was a lamp used, or natural light? 

Response: To ensure consistency in light intensity, the light source of the photosynthesis instrument was used, with a light intensity set at 1200 μmol·m-2·s-1 (Line: 433-435).

  1. The Ci values as high as 410 umol mol-1 suggest high CO2 during the measurements. 

Response: Thanks so much for alerting us to the problem. After inspection, it was found that Ci and Gs were labeled upside down in Figure 3. We have conducted a comprehensive inspection and revision (see Figure 3).

  1. A careful description of how measured leaves were chosen is also needed - where in the canopy?  It seems like the sampling of leaves for epidermal structure differed from that for SLW, and the leaf gas exchange sampling is thus totally unclear.  

Response: We have rewritten the sampling method (Line: 389-393). In fact, we built a ten-meter-high platform to collect samples and measure photosynthesis of leaves from the standard treetops of the canopy (Line: 431-433).

Round 2

Reviewer 2 Report

Comments and Suggestions for Authors

It looks to me like the leaf gas exchange measurements have substantial errors.  Possibly due to unstable CO2?  The problem is clear in the poor correlation between Gs and Tr despite only small variation in VPD.  This does not make sense.  Also the Ci values do not make sense in relation to the Pn and Gs values.  I think that the leaf gas exchange measurements should be eliminated from the paper.

Although specific leaf weight (SLW) is mentioned in the text, I do not see any data about it.  Leaf nutrient contents make much more sense if expressed as nutrients per unit of area rather than per unit of mass.  In any event, we need to be given the data on SLW, and also on mean area per leaf at the different densities. 

Author Response

RESPONSE TO COMMENTS

Reviewer #2:

  1. It looks to me like the leaf gas exchange measurements have substantial errors. Possibly due to unstable CO2? The problem is clear in the poor correlation between Gs and Tr despite only small variation in VPD.  This does not make sense.  Also the Ci values do not make sense in relation to the Pn and Gs values.  I think that the leaf gas exchange measurements should be eliminated from the paper.

Response: We fully agree with the reviewer's point of view that based on the data obtained through a portable photosynthesis system, it is hard to infer the results related to leaf gas exchange. Therefore, in the discussion section, we have removed the relevant statements.

  1. Although specific leaf weight (SLW) is mentioned in the text, I do not see any data about it. Leaf nutrient contents make much more sense if expressed as nutrients per unit of area rather than per unit of mass. In any event, we need to be given the data on SLW, and also on mean area per leaf at the different densities.

Response: According to the suggestions, an analysis of the differences in leaf area and specific leaf area under varying stand density was added (Figure 2c-d). We found that the variation trends of leaf area and specific leaf area in samples with different densities were consistent, roughly showing an increasing trend with increasing density (Line: 135-137). Perhaps due to the high correlation between specific leaf area and light level, which decreases with increasing light intensity, so trees grown under low-density planting usually have thicker leaves with lower specific leaf area (Line: 320-322).

Round 3

Reviewer 2 Report

Comments and Suggestions for Authors

I thought the authors would remove the table on leaf gas exchange, since they admitted that it was not reliable.  This was user error: that type instrument is reliable if used correctly.

I appreciate the new inclusion of leaf mass per area data.

Author Response

Dear reviewer,

According to the reviewer's suggestion, the content related to "leaf gas exchange" has been deleted throughout the current version of the paper. In Table 1, the content of " leaf gas exchange" was not found. If relevant content appears elsewhere, please provide the "line number" to help us make modifications. Once again, thank you to the reviewer for your valuable suggestions on the paper's revisions.

Ying Guo